# Release of Exosomal PD-L1 in Bone and Soft Tissue Sarcomas and Its Relationship to Radiotherapy

**DOI:** 10.3390/cancers16132489

**Published:** 2024-07-08

**Authors:** Keisuke Yoshida, Kunihiro Asanuma, Yumi Matsuyama, Takayuki Okamoto, Tomohito Hagi, Tomoki Nakamura, Akihiro Sudo

**Affiliations:** 1Department of Orthopaedic Surgery, Mie University Graduate School of Medicine, Tsu 514-8507, Mie, Japan; keeesuke1224@gmail.com (K.Y.); m-yumi@clin.medic.mie-u.ac.jp (Y.M.); hagifana@clin.medic.mie-u.ac.jp (T.H.); tomoki66@clin.medic.mie-u.ac.jp (T.N.); a-sudou@clin.medic.mie-u.ac.jp (A.S.); 2Department of Pharmacology, Faculty of Medicine, Shimane University, Izumo 693-0021, Shimane, Japan; okamoto@med.shimane-u.ac.jp

**Keywords:** programmed death-ligand 1, soft tissue, bone, sarcoma, exosome

## Abstract

**Simple Summary:**

Programmed death-ligand 1 (PD-L1) on the tumor cell surface binds to its receptor, programmed death-1 (PD-1), on T cells and inhibits their activity. Immunotherapy involving PD-L1 and PD-1 inhibition has provided a remarkable innovation in cancer therapy. Recently, the soluble form of PD-L1 (sPD-L1) has attracted attention for systemic immune suppression. In this study, we successfully showed that sarcoma cells can release functional exosomal PD-L1, one of the forms of sPD-L1. Furthermore, the release of exosomal PD-L1 derived from normal and sarcoma cells was shown to be induced by irradiation. These findings demonstrate that radiated cells, including normal cells and sarcoma cells, induce the systemic release of exosomal PD-L1, and combination therapy with anti-PD-1/PD-L1 antibody may block the immune activity of exosomal PD-L1.

**Abstract:**

(1) Background: Exosomal PD-L1 has garnered attention owing to its role in instigating systemic immune suppression. The objective of this study is to elucidate whether bone and soft tissue sarcoma cells possess the capacity to secrete functionally active exosomal PD-L1 and whether radiotherapy (RT) induces the exosomal PD-L1 release. (2) Methods: Human osteosarcoma cell line 143B and human fibrosarcoma cell line HT1080 were utilized. Exosomes were isolated from the culture medium and blood via ultracentrifugation. The expression of PD-L1 on both tumor cells and exosomes was evaluated. The inhibitory effect on PBMC was employed to assess the activity of exosomal PD-L1. Post radiotherapy, changes in PD-L1 expression were compared. (3) Results: Exosomal PD-L1 was detected in the culture medium of tumor cells but was absent in the culture medium of PD-L1 knockout cells. Exosomal PD-L1 exhibited an inhibitory effect on PBMC activation. In tumor-bearing mice, human-derived exosomal PD-L1 was detected in the bloodstream. Following radiotherapy, tumor cells upregulated PD-L1, and human-derived exosomal PD-L1 were detected in the bloodstream. (4) Conclusions: Exosomal PD-L1 emanates from bone and soft tissue sarcoma cells and is disseminated into the circulatory system. The levels of PD-L1 in tumor cells and the release of exosomal PD-L1 were augmented after irradiation with RT.

## 1. Introduction

In the microenvironment of cancer, innate immunity is inhibited in a process known as immune tolerance. One of the mechanisms of immune tolerance is caused by immune checkpoint proteins (ICPs). ICP inhibitors are a new form of systemic therapy. They “release the brakes” on the immune system, allowing the enhanced ability of T cells to specifically target and kill cancer cells. Programmed death receptor ligand 1 (PD-L1), one of the ICP molecules, is a membrane-bound ligand found on the cell surface of many cell types that are upregulated in inflammation and some oncogenic lesions [1]. Programmed death receptor 1 (PD-1) is a receptor on T cells that sends a suppressive signal that inhibits T cell anti-neoplastic activity by binding to PD-L1 [2]. Normal cells are believed to express PD-L1 in an inflammatory environment, suppress T cells, and prevent excessive tissue damage from long-term persistence and the spread of inflammation [3]. However, cancer has been implicated in preventing attacks from the immune system by suppressing T cell activation by binding the PD-L1 expressed on cancer cells to the PD-1 on cytotoxic T cells [4]. When ICP inhibitors, such as antibodies against PD-L1 and PD-1, “release the brakes” and are allowed to react to a specified antigen, PD-1 is unable to transmit PD-L1 signals, and an anti-tumor effect is exerted [5]. Melanoma, non-small-cell lung cancer (NSCLC), and renal cancer have high PD-L1 levels on the tumor cell membrane (cPD-L1), and anti-PD-1 or anti-PD-L1 antibodies have shown effectiveness against these cancers [6]. Conversely, the positive rate of cPD-L1 expression ranges widely, from 2.5% to 64.8% [7,8,9,10]. Tumor-infiltrating PD1-positive lymphocytes and PD-L1 expression predict a poor prognosis of soft tissue sarcomas [10]. Regarding the therapeutic effect of anti-PD-1 antibodies against bone and soft tissue sarcoma, in an interim report on a phase 2 trial, only 18% of soft tissue sarcomas and 5% of osteosarcoma patients had a response [11]. Anti-PD-1 therapy efficacy in bone and soft tissue sarcomas is controversial. Bone and soft tissue tumors may be less associated with cPD-L1.

Extracellular vesicles (EVs), also known as microparticles, are tiny, membrane-enclosed sacs that are thought to be shed from the surface of healthy or damaged cells under conditions such as cell activation, growth, and apoptosis. A particular form of EVs is the exosome, derived from the endocytic pathway. Recent studies have shown that PD-L1 is found on the surface of exosomes, and PD-L1 expressed on the exosomes may bind to PD-1 and inhibit the anti-tumor immune response [12]. Melanoma cells are considered to release exosomal PD-L1 into the tumor microenvironment and circulation to battle anti-tumor immunity systemically [13]. Recent studies suggest the presence of exosomal PD-L1 isolated from the blood samples of cancer patients, and the level of PD-L1 correlates with the patients’ pathological features [14,15,16]. Furthermore, there has been a focus on the combination of radiotherapy (RT) and anti-PD-1/PD-L1 antibody therapy. RT is known to induce cPD-L1 upregulation, and combination therapy with anti-PD-1/PD-L1 has significantly improved the clinical outcomes of various cancers [17,18]. In our previous study, the concentration of the soluble form of PD-L1 (sPD-L1) could predict future metastasis and prognosis in soft tissue sarcoma patients better than cPD-L1 [19]. The sPD-L1 includes several types of soluble forms, and exosomal PD-L1 is a major feature. Given this, we hypothesized that exosomal PD-L1 is released from bone and soft tissue sarcoma cells, exerting an inhibitory influence on immune cells. The purpose of this study is to elucidate whether bone and soft tissue sarcoma cells are capable of secreting exosomal PD-L1, and further, to ascertain the biological activity of exosomal PD-L1. Additionally, we have explored whether RT induces the release of exosomal PD-L1.

## 2. Materials and Methods

### 2.1. Cell Lines

Human osteosarcoma cell line 143B and human fibrosarcoma cell line HT1080 were cultured in Dulbecco’s modified Eagle’s medium (DMEM) containing 5% fetal bovine serum (FBS). All cell cultures were maintained in 5% CO_2_ at 37 °C.

### 2.2. Exosome Isolation

Exosomes were extracted in vitro by ultracentrifugation from the culture medium [20]. 143B and HT1080 cells grown to 70% confluence were washed twice with phosphate-buffered saline (PBS) and then grown in 25 mL serum-free Opti-MEM medium. After 48 h of incubation, the conditioned medium was collected and centrifuged at 3000× *g* for 15 min at 4 °C to thoroughly remove cellular debris. The supernatants were ultracentrifuged at 100,000× *g* for 70 min at 4 °C. The pellets were washed with PBS, ultracentrifuged, and resuspended in 10 μL PBS. In vivo, exosomes were extracted from serum using blood collected from mice. Serum samples were ultracentrifuged using the same method. Isolated exosomes were stored at –80 °C until needed.

### 2.3. Measurement of Interferon Gamma (IFN-γ) Production

Human peripheral blood mononuclear cells (PBMCs) were isolated using a Ficoll gradient (cytiva, Tokyo, Japan). The resulting cells were pretreated for 30 min with exosomes isolated from HT1080 cells in the presence of anti-human PD-L1 (Merck Millipore, Burlington, MA, USA, cat. MABC980) or control antibody (BioLegend, San Diego, CA, USA, cat. 400301), followed by incubation with phorbol 12-myristate-13-acetate (PMA, Tokyo, Japan; 50 ng/mL, AdipoGen Corp, San Diego, CA, USA, cat. AG-CN2-0010) and ionomycin (500 ng/mL, AdipoGen Corp, cat. AG-CN2-0416). After 4 h, an enzyme-linked immunosorbent assay was performed using human IFN-γ (Human IFN-gamma Sandwich ELISA Kit, Proteintech, Tokyo, Japan, cat. KE00146), according to the manufacturer’s protocol. We prepared n = 5 of each data and statistically analyzed them.

### 2.4. Flow Cytometry Analysis

The cells were incubated with anti-human PD-L1 (BioLegend, cat. 329705) and isotype control antibodies (BioLegend, cat. 981804) for 30 min at room temperature. All analyses were performed using a FACS scan flow cytometer (BD Biosciences, Franklin Lakes, NJ, USA). In addition, data were analyzed with C6 software ver1.0.264.21 (BD Biosciences, Franklin Lakes, NJ, USA).

### 2.5. Immunofluorescence Staining

Immunofluorescence staining was performed on formalin-fixed, paraffin-embedded (FFPE) sections using 4% PFA. Fixed cells were permeabilized with 0.1% Triton X-100 and blocked with bovine serum albumin buffer for 1 h. For FFPE sections, antigen activation was performed by steeping the sections in citrate buffer (pH = 6.0) before blocking. FFPE sections were incubated with the primary antibody overnight at 4 °C, followed by incubation with the fluorescent dye-conjugated secondary antibody for 1 h. The nuclei were stained with DAPI (Thermo Fisher Scientific, Waltham, MA, USA, cat. 62248). The samples were observed under a Nikon confocal microscope at 100× magnification.

### 2.6. Western Blot Analysis

Whole cell lysates or exosomal proteins were separated by 12% sodium dodecyl sulfate–polyacrylamide gel electrophoresis and transferred to nitrocellulose membranes. Blots were blocked using 5% nonfat dry milk for 1 h at room temperature and incubated overnight at 4 °C with the corresponding primary antibody at the dilution recommended by the supplier, followed by incubation with an HRP-labeled secondary antibody (Dako, Santa Clara, CA, USA, cat. P0448) for 1 h at room temperature. Blots on the membrane were developed using ECL detection reagent (Pierce, Appleton, WI, USA). We used an anti-human PD-L1 rabbit monoclonal antibody (Cell Signaling Technology, Danvers, MA, USA, cat. 13684, E1L3N) and an anti-mouse PD-L1 rabbit monoclonal antibody (GeneTex, Irvine, CA, USA, cat. GTX638348). CD9 (Santa Cruz Biotechnology, Dallas, TX, USA, cat. sc-9148), CD63 (Santa Cruz Biotechnology, cat. sc-15363), and CD81 (Santa Cruz Biotechnology, cat. sc-9158) were used as exosome markers, and actin (Santa Cruz Biotechnology, cat. sc-47778) was used as the loading control. The intensity of each band was compared using ImageJ software ver1.54g.

### 2.7. PD-L1 Knockout HT1080 Cells/CRISPR-Cas9

The CRISPR-Cas9 system was used to generate PD-L1 knockout HT1080 cells. HT1080 cells were seeded in 24-well cell culture plates at a density of 1 × 10^4^ cells/cm^2^ in 500 µL of medium. The next day, 1250 ng of Cas9 protein (TrueCut Cas9 protein v2, Invitrogen, Carlsbad, CA, USA) with 240 ng of gRNA (GGTTCCCAAGGACCTATATG, TrueGuide™ Synthetic sgRNA, A35533, Invitrogen) and 2.5 µL of lipofectamine CRISPRMAX^®^ (Invitrogen) were added to 25 µL of Opti-MEM medium (Invitrogen) at room temperature (Tube 1). Lipofectamine CRISPRMAX^®^ solution was added to 25 µL of Opti-MEM medium and incubated for approximately 1 min at room temperature (Tube 2). Tube 2 was added to Tube 1, mixed by pipetting, and incubated at room temperature for 10–15 min. This mixed transfection complex reagent was added to cells and incubated for 2 days. Cells were resuspended in a 96-well plate by a limited dilution method. The isolated cells were cultivated and expanded into a Petri dish to reach 80% confluence. Cells were collected, and PD-L1 expression was confirmed by Western blot. PD-L1 knockout cells were selected. The proliferation rate of the selected cells was confirmed by the MTS assay (Promega, Tokyo, Japan, cat. G3581) at 24 h and 48 h; cPD-L1 and exosomal PD-L1 were confirmed by Western blotting and FCM.

### 2.8. Experiments with Mice

Male BALB/c nude mice (8 weeks old) were purchased from Clair (Tokyo, Japan) to establish the xenograft model. Tumor cells (1 × 10^6^ cells per 0.1 mL) were injected subcutaneously into the back. When the tumors had grown to about 5 mm in diameter, the mice were irradiated with 12 Gy, and exosomes were isolated from the blood of each mouse 1 to 5 days after irradiation. PD-L1 expression was evaluated using Western blotting.

### 2.9. Statistical Analysis

Data are presented as mean ± standard deviation values. *p*-values were determined using the ANOVA method, which compares mean values among three or more groups.

## 3. Results

### 3.1. PD-L1 Expression in Bone and Soft Tissue Sarcoma Cell Lines

To demonstrate PD-L1 expression in tumor cells, immunostaining, Western blot, and flow cytometry were performed. On immunostaining analysis, PD-L1 (green) was observed in each HT1080 cell (left) and each 143B (right) cell. No staining was seen with rabbit IgG or without primary antibody (Figure 1A,B). On Western blotting, the PD-L1 band was thicker in HT1080 cells than in 143B cells. The original western blot figure can be found in Appendix A. On FCM, immunofluorescence for PD-L1 was shifted more in HT1080 cells than in 143B cells (Figure 1C). These results indicated that PD-L1 was markedly expressed by HT1080 cells and slightly by 143B cells.

### 3.2. Exosomal PD-L1 Expression in Culture Medium and Serum

Exosomes in the culture medium were extracted by ultracentrifugation, and Western blotting was performed. In HT1080 cells, exosome markers CD9, CD63, and CD81 were detected, and PD-L1 was highly expressed (Appendix A). In 143B cells, exosome markers CD9, CD63, and CD81 were detected (Appendix A). PD-L1 expression was observed in the exosome-containing fraction of HT1080 and 143B cells.

BALB/c nude mice were transplanted with HT1080 cells or 143B cells, and blood was obtained when tumors grew to about 10 mm in diameter. Exosome fractions were extracted by ultracentrifugation of serum. Exosome markers in sera from control healthy mice (no tumor cell transplantation) and human PD-L1 were not detectable (Appendix A). When mice were implanted with HT1080 cells or 143B cells, CD63-positive exosomes and human PD-L1 expression were detected (Appendix A).

In addition, the anti-human PD-L1 antibody did not react with PD-L1 from mouse osteosarcoma cell line LM8 (Appendix A), suggesting that circulating exosomal PD-L1 was released from transplanted tumor cells. Western blotting of mouse-derived PD-L1 was not performed because anti-mouse PD-L1 antibodies cannot distinguish between human and mouse PD-L1 (Appendix A).

To confirm non-specific reaction by secondary antibody, Western blotting was performed using a secondary antibody. Non-specific bands were not observed (Appendix A).

### 3.3. Inhibitory Effect of Exosomal PD-L1 on T Cell Activation In Vitro

PBMCs were activated by PMA and ionomycin and released IFN-γ. However, adding exosomes from HT1080 cells successfully inhibited IFN-γ release. Furthermore, the addition of an anti-human PD-L1 antibody blocked suppression of IFN-γ release by exosomes, resulting in higher IFN-γ release. On the other hand, when control antibodies of the same class (BioLegend, cat. 400301) were added along with exosomes, a blocking effect as with the anti-human PD-L1 antibody was not observed, and the release of IFN-γ was suppressed. These findings suggest that exosomes containing PD-L1 inhibited T cell activation (Figure 2). The standard curve (Appendix A) and data for each n = 5 (Appendix A) of ELISA results were included as Appendix A.

### 3.4. PD-L1 Knockout (KO) HT1080 Cells/CRISPR-Cas9

Four clones (A-D) exhibiting exceptional proliferation were chosen for analysis. FCM confirmed the expression of cPD-L1 in HT1080 cells (Figure 1C), whereas none of the four KO cells expressed cPD-L1 (Figure 3A–D). Western blotting also confirmed the expression of cPD-L1 in HT1080 cells, whereas none of the four KO cells expressed cPD-L1 (Appendix A). In the MTS assay, KO cells showed various proliferation rates compared to wild HT1080 cells (Figure 4). Western blotting demonstrated that the culture medium of these KO cells showed the presence of the exosome marker, CD81, though no expression of exosomal PD-L1 was observed (Appendix A).

As Appendix A, Western blotting was performed using these samples and secondary antibody, but there was no reaction (Appendix A).

### 3.5. Effect of Irradiation on Exosomal PD-L1 Protein Levels In Vitro

To examine whether irradiation to tumor cells induced exosomal PD-L1 release, the exosomal fractions from the culture media of irradiated HT1080 cells and 143B cells were investigated. When the radiation doses were set to 0, 3, 6, and 9 Gy, 6 Gy irradiation tended to increase the expression of cPD-L1 in HT1080 cells at 24 and 48 h (Appendix A). In 143B cells, the cPD-L1 expression levels tended to be increased in a radiation intensity-dependent manner 48 h after irradiation (Appendix A). In contrast, exosomes obtained from the culture medium of irradiated HT1080 cells showed an intensity-dependent tendency to increase in exosomal PD-L1 protein levels at 48 h after irradiation (Appendix A). A similar study of 143B cells showed that exosomal PD-L1 levels tended to be increased 24 h after 6 Gy irradiation (Appendix A).

### 3.6. Effect of Irradiation on Exosomal PD-L1 Protein Levels In Vivo

The biological effects of irradiation on cellular and exosomal PD-L1 expressions were investigated. First, the effect of irradiation on healthy control mice was evaluated. Circulating exosomal PD-L1 (anti-mouse) expression (46 kDa) was maximal three days after irradiation (Appendix A). As Appendix A, Western blotting was performed using these samples and a secondary antibody, but there was no reaction (Appendix A).

This indicated that irradiation induced exosomal PD-L1 release from normal cells. Next, after the tumor grew to about 5 mm in diameter, the mice were irradiated with 12 Gy. In tumor tissues, cellular PD-L1 expression tended to be upregulated two days after irradiation, followed by a tendency to decrease in PD-L1 protein levels (Figure 5A). In contrast, irradiation to 143B tumor cells did not show significant histological upregulation of cellular PD-L1 (Figure 5B).

In addition, circulating exosomal PD-L1 was analyzed. In HT1080 cells, the expression of exosomal PD-L1 detected by the anti-human PD-L1 antibody strongly tended to be enhanced five days after irradiation (Appendix A). In contrast, no human-derived exosomal PD-L1 expression was observed in 143B cells after irradiation (Appendix A). This may be due to the amount of PD-L1 expression from 143B tumor cells.

Western blotting of mouse-derived PD-L1 was not performed because mouse antibodies cannot distinguish between human and mouse PD-L1 (Appendix A). As Appendix A, Western blotting was performed using these samples and a secondary antibody, but there was no reaction (Appendix A).

## 4. Discussion

In recent years, sPD-L1 has been identified in the blood of cancer patients [21,22,23], and many studies have been conducted to investigate its significance. In general, elevated levels of sPD-L1 in cancer patients have been shown to correlate with a poor prognosis or resistance to therapy. For instance, high levels of sPD-L1 were also observed in patients diagnosed with metastatic clear-cell renal-cell carcinoma compared to those with non-metastatic disease [24], and sPD-L1 levels were significantly elevated in patients with muscle-invasive and metastatic bladder cancer [25]. Similarly, high levels of sPD-L1 have been associated with poor prognoses in patients with hepatocellular carcinoma [26], melanoma [27], and small-cell lung cancer [28]. In addition, we previously reported that a high level of sPD-L1 predicted future metastasis and a poor prognosis in soft tissue sarcoma patients [17].

Exosomes facilitate the transfer of their contents, such as nucleic acids, lipids, and proteins, to target cells, thereby mediating intercellular signaling, drug resistance, and immunomodulation [29,30]. Exosomal PD-L1, one form of sPD-L1, can transport bioactive PD-L1, which binds to PD-1 on the surface of T cells, delivering inhibitory signals [13,17]. In addition, exosomal PD-L1 has been found in various studies to promote immunosuppression and the growth of multiple tumors. Replacing PD-L1 exogenously expressed on Raji B cells with exosomal PD-L1 derived from PC3 cells inhibited Jurkat T cell activation and reduced IL-2 secretion [12]. Similarly, tumor-derived exosomal PD-L1 inhibited T cell activation in draining lymph nodes in vivo, and exogenous exosomal PD-L1 was found to promote tumor growth [12]. In NSCLC patients, exosomal PD-L1 inhibited CD8+ T cell activity dose-dependently, induced CD8+ T cell apoptosis, and decreased IL-2 and IFN-γ production [31]. In the present study, exosomes derived from sarcoma cells inhibited IFN-γ production. This inhibitory effect was inactivated by the anti-PD-L1 antibody. It was successfully shown that sarcoma cells can release functional exosomal PD-L1. Clinically, exosomal PD-L1 has also been reported to be associated with prognosis. For example, high levels of exosomal PD-L1 were correlated with advanced tumor stage, larger tumor size (>2.5 cm), lymph node metastasis, and distant metastasis in patients with NSCLC [32]. In addition, exosomal PD-L1 has been associated with poor prognosis in patients with pancreatic ductal adenocarcinoma [33]. In our research, exosomal PD-L1 derived from sarcomas implanted in mice was observed in blood, and we successfully demonstrated that sarcomas can be a supplier of circulating exosomal PD-L1 in vivo. This indicated that sarcomas evade the systemic immune system by releasing exosomal PD-L1.

RT is a primary form of cancer treatment used to treat various types of cancer, regardless of clinical stage. Recently, combination therapy of RT and anti-PD-1/PD-L1 antibodies has attracted significant attention. Several studies have demonstrated immune activation by irradiation. Specifically, it has been reported that RT activates host immunity induced by immunogenic cell death [34], induces IFN-γ that leaves dendritic cells, and stimulates innate immunity associated with IFN pathway activity [35], as well as inducing chemokines and adhesion factors involved in T cell migration and infiltration [36]. These findings suggest that radiation-induced immunomodulation mechanisms function in various phases of the cancer immune cycle.

On the other hand, RT attracts immunosuppressive cells such as M2 tumor-associated macrophages, myeloid-derived suppressor cells, regulatory T cells, and N2 neutrophils, which have immunosuppressive effects in the tumor microenvironment by releasing immunosuppressive cytokines (TGF-β, IL-10) and chemokines [37]. Furthermore, several preclinical studies have reported irradiation-induced PD-L1 expression [18,19,38,39]. The IFN-dependent pathway is considered a common pathway for expressing PD-L1, and it is also responsible for the upregulation of PD-L1 after irradiation. The induction of PD-L1 by IFN-γ is more persistent and more substantial through the JAK-STAT-IRF pathway [40]. In addition, recent studies have demonstrated that activation of ATM/ATR/Chk1 kinase in response to DNA double-strand breaks by irradiation can enhance PD-L1 expression in cancer cells [41].

Regarding the two aspects of RT function, i.e., immunosuppression and immune activation, anti-PD-1/PD-L1 antibodies can alleviate the RT-associated immunosuppression and therefore make sense as an adjunctive therapy with RT.

In recent studies, combining RT and anti-PD-1/PD-L1 antibodies improved the immune response. RT in combination with anti-PD-1/PD-L1 antibodies for carcinoma-bearing mice produced an abscopal effect that inhibited the growth of unirradiated tumors rechallenged on the opposite flank, suggesting the induction of a systemic, sustained anti-tumor immune response [42]. In clinical trials of combination therapy, the phase III Pacific trial, conducted in patients with locally advanced unresectable NSCLC, has significantly impacted the field of clinical oncology. Administering durvalumab as consolidation therapy within 1–42 days after concurrent chemoradiotherapy significantly extended progression-free survival compared to the placebo [43]. Furthermore, a subgroup analysis showed that the overall survival of patients who received durvalumab within 1–14 days after completing chemoradiotherapy was also significantly prolonged [44]. Thus, RT with anti-PD-1/PD-L1 antibody treatment may sustain host immunity in a favorable immune environment.

However, there is considerable debate regarding whether the prolonged prognosis is only a local reaction between PD-L1 on tumor cells and PD-1 on immune cells. The present study showed that irradiation of HT1080 cells and 143B cells increased not only cPD-L1 expression of tumor cells but also the release of exosomal PD-L1 in the culture medium. Furthermore, irradiation to control mice increased exosomal PD-L1 derived from normal cells, and irradiation to tumor-bearing mice enhanced exosomal PD-L1 release derived from normal cells of mice and implanted tumor cells. These findings suggest that radiated cells including normal cells and tumor cells induce systemic release of exosomal PD-L1, which leads to the risk of exacerbation of distant lesions by systemic immune suppression via exosomal PD-L1. Based on this notion, excessive wide radiation has a potential systemic disadvantage, and localized radiation therapy, such as stereotactic body radiation therapy (SBRT), is preferred. Furthermore, combination therapy with an anti-PD-1/PD-L1 antibody may block immune suppression from cellular PD-L1 and exosomal PD-L1.

## 5. Conclusions

Exosomal PD-L1 is released from bone and soft tissue sarcoma cells and subsequently enters the circulatory system. The cPD-L1 level and release of exosomal PD-L1 increased after RT irradiation. Localized radiation therapy is preferred to avoid inducing excessive cPD-L1 expression and exosomal PD-L1 production from normal cells. The findings suggest that further studies are needed to determine whether combining RT and PD-1/L1 inhibitors for bone and soft tissue sarcomas can enhance the anticancer therapeutic effect.

## Figures and Tables

**Figure 1 cancers-16-02489-f001:**
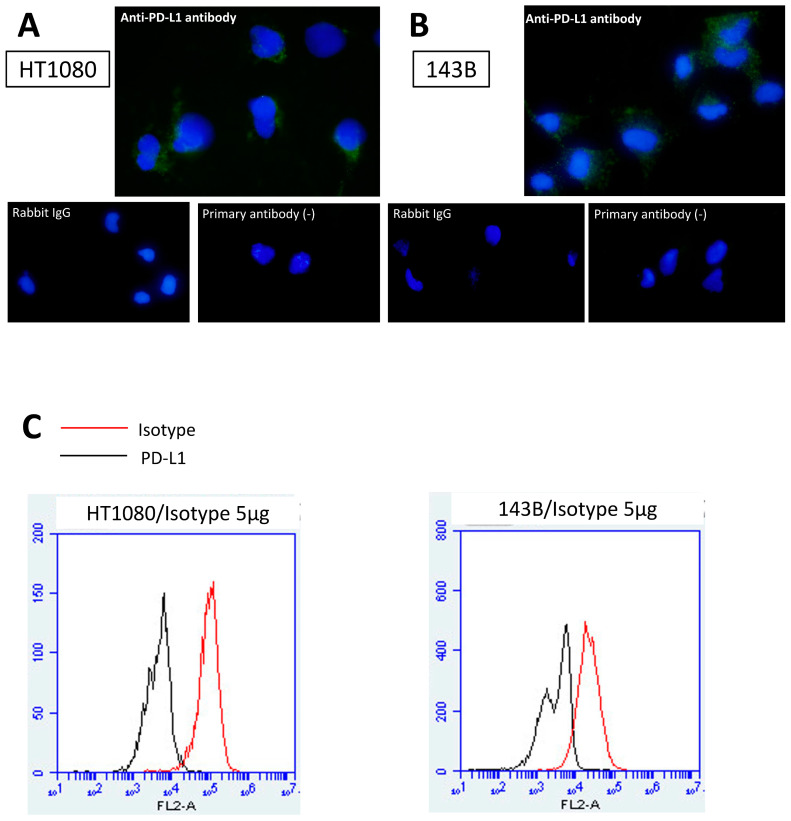
PD-L1 expression in HT1080 cells and 143B cells. Immunocytochemistry for HT1080 (**A**) and 143B (**B**) cells, and flow cytometry (**C**) were performed.

**Figure 2 cancers-16-02489-f002:**
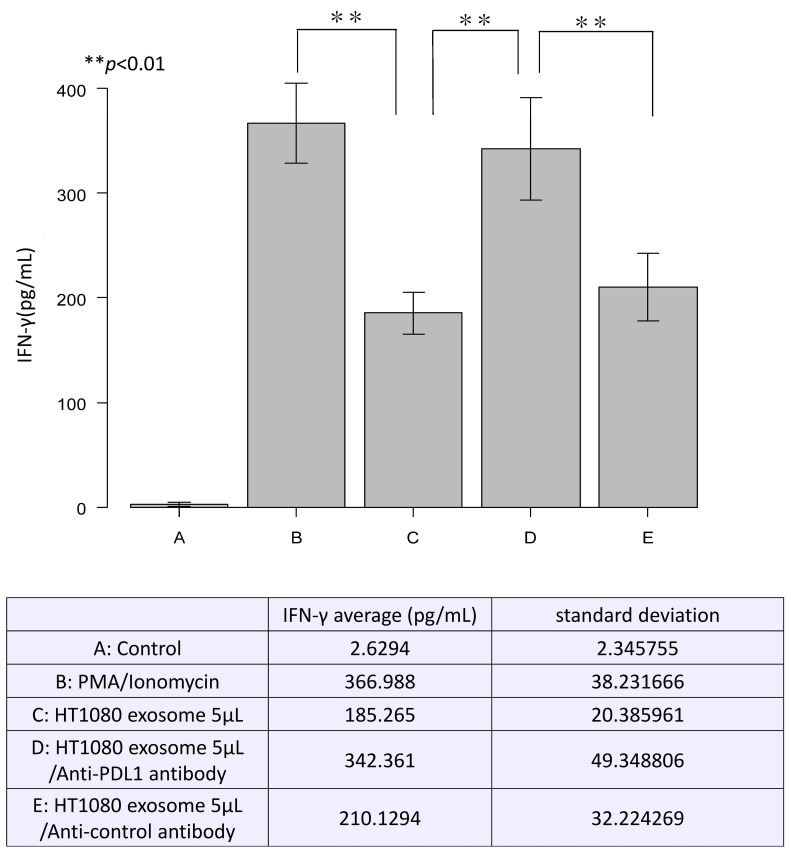
Effect of PD-L1-containing exosomes on activated human peripheral blood mononuclear cells (PBMCs). PBMCs were activated with PMA and ionomycin, and the amount of IFN-γ was measured as a reference of activation. Exosomes isolated from the culture medium of HT1080 cells were used. PD-L1 antibody or a class-matched control antibody was added to examine the specificity of PD-L1 on exosomes. These data were prepared n = 5 each and statistically analyzed using the ANOVA method.

**Figure 3 cancers-16-02489-f003:**
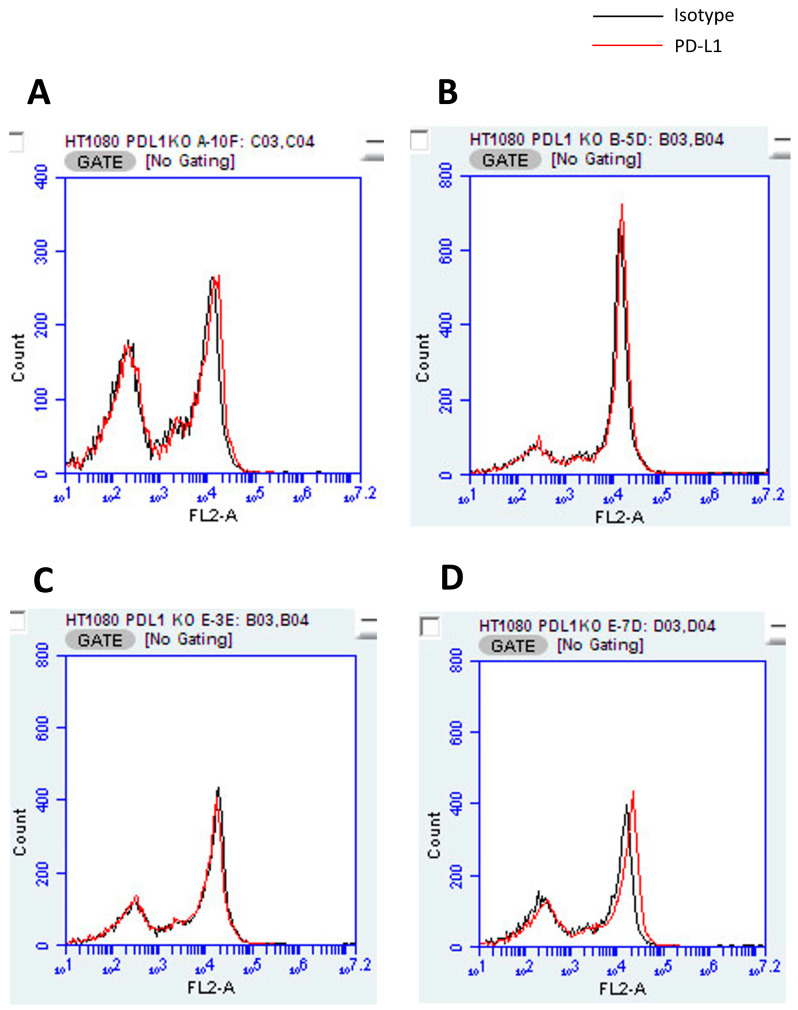
PD-L1 knockout HT1080 tumor cells according to flow cytometry. PD-L1 knockout cells were evaluated by flow cytometry. Flow cytometry confirms the expression of cPD-L1 in HT1080 cells, whereas none of the four cells (**A**–**D**) shows cPD-L1 expression.

**Figure 4 cancers-16-02489-f004:**
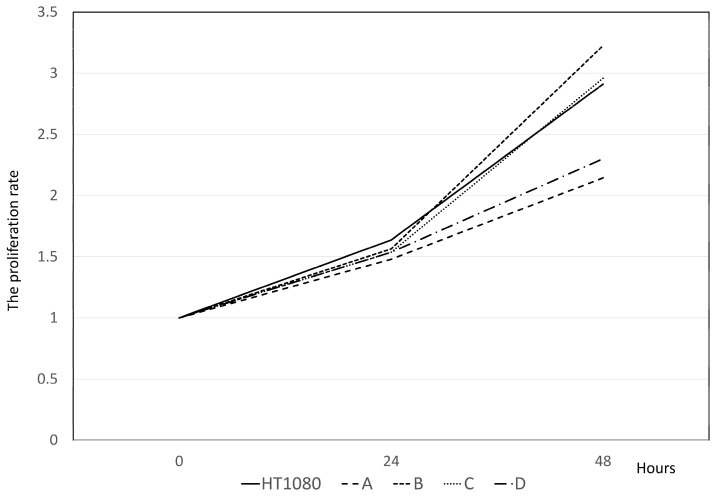
The proliferation rate of HT1080 and cells A–D in the MTS assay. The cell proliferation rate of the knockout cells A–D was evaluated by MTS assay. The cell proliferation rate of these cells is also confirmed by MTS assay. The proliferation rate of C cells closely resembles that of normal HT1080 cells after 48 h of incubation.

**Figure 5 cancers-16-02489-f005:**
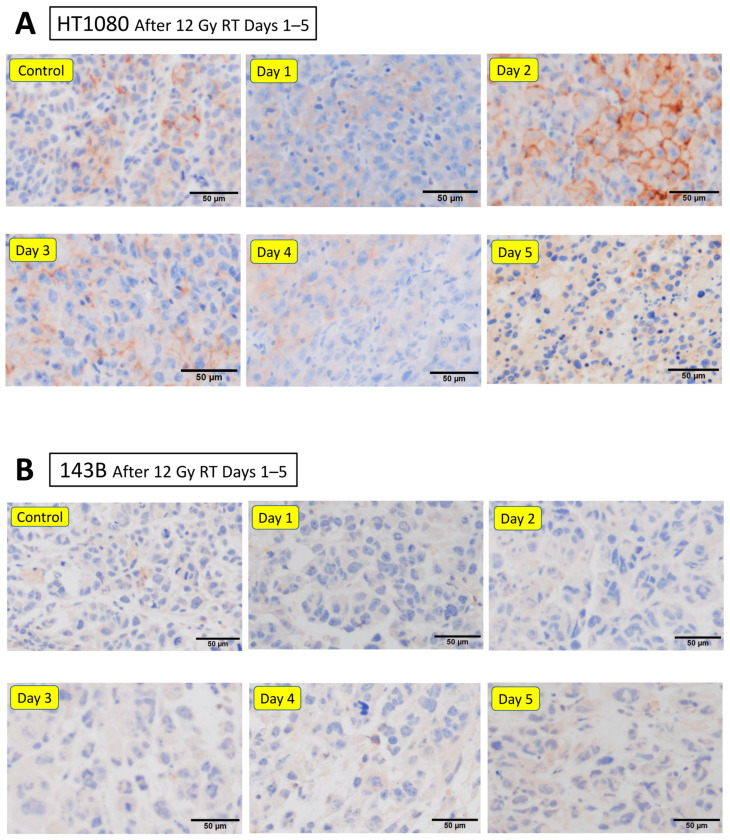
Effects of irradiation on PD-L1 expression on immunohistochemistry. Immunohistochemical findings of implanted tumor using anti-human PD-L1 antibody after radiation are shown. In HT1080 cells, cellular PD-L1 expression tends to be upregulated two days after irradiation (**A**). Irradiation to 143B tumor cells does not show histological upregulation of cellular PD-L1 (**B**).

## Data Availability

The data presented in this study are available in this article and Appendix A.

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
