# Peer review of "Release of Exosomal PD-L1 in Bone and Soft Tissue Sarcomas and Its Relationship to Radiotherapy"

_cancers, 2024, doi:10.3390/cancers16132489_

Round 1

Reviewer 1 Report

Comments and Suggestions for Authors

The manuscript of Yoshida et al. is proposed for analyzing how exosomal PD-L1 is released from bone and soft tissue sarcoma cells, and if it exerts an inhibitory influence on immune cells.

Exosomes were isolated from the culture medium of osteosarcoma cell line 143B and fibrosarcoma cell line HT1080 and from blood of BALB/c nude mice injected with cells.

According to the authors, exosomal PD-L1 was detected in the culture medium of tumor cells, but absent in the culture medium of PD-L1 knockout cells.

In tumor-bearing mice, human-derived exosomal PD-L1 was detected in the bloodstream.

Following radiotherapy, tumor cells upregulated PD-L1, and both mouse and human-derived exosomal PD-L1 were detected in the bloodstream.

The strategy of the author is very well defined and fit well with the goal of the study.

The results are well organized but a major problem is observed regarding Western blotting analysis.

When working with exosome, one of the main challenge is exosome purification and characterization. In this study, exosome purification was done by ultracentrifugation and characterization was done by WB with exosome specific markers.

The risk with such protocol of purification is to keep large cellular membrane in addition of exosomes. How to be sure that you analyzed only exosome and not cellular debris and membrane?

Figure 2: the WB pictures appears not very well defined with a lot of background on the top.

Please add the molecular weight markers on the left.

Quantify the bands for each antibody when needed (if you explain an increase of expression)

In this figure, CD9, CD63 and CD81 are not convincing, please provide better pictures or use other exosome markers

All figures with WB:

Regarding the bands cut for each antibody, please provide the whole (uncut) membrane pictures with the corresponding MW markers for all analysis

Author Response

Reviewer 1

Comment 1

The risk with such protocol of purification is to keep large cellular membrane in addition of exosomes. How to be sure that you analyzed only exosome and not cellular debris and membrane?

Rersponse 1

We are very grateful to the reviewer’s critical comments. The sPD-L1 includes several types of soluble forms, including exosome, transmembrane domain-deficient variant, and cell debris. In the previous study about exosomal PD-L1, soluble PD-L1 by ultracentrifugal extraction mainly consist of exosome(Cell. 2019 April 04; 177(2): 414–427.e13. doi:10.1016/j.cell.2019.02.016.). Therefore, we chose the ultracentrifugal extraction method.

Comment 2

Figure 2: the WB pictures appears not very well defined with a lot of background on the top.

Please add the molecular weight markers on the left. Quantify the bands for each antibody when needed (if you explain an increase of expression)

In this figure, CD9, CD63 and CD81 are not convincing, please provide better pictures or use other exosome markers

Response 2

Thank you for your pointing. We added molecular weight markers in all WB pictures. We agree with your assessment for Figure 2. We performed WB in Figure 2 several times, but black shadows inevitably appeared. We have no idea about these shadows. Our laboratory has only CD9, CD63 and CD81 as exosome markers. We do not have enough time to perform another western blot after purchasing new antibody until dead end date.

Comment 3

All figures with WB:

Regarding the bands cut for each antibody, please provide the whole (uncut) membrane pictures with the corresponding MW markers for all analysis

Response 3

Thank you for your pointing. We added new figures with WB and uncut pictures as supplemented figures. (Figure 2,3,6,8,9,10,12)

Reviewer 2 Report

Comments and Suggestions for Authors

This manuscripts reports on studies investigating the production of exosomal PD-L1 produced by two human tumor cells lines before and after radiation treatment.  A few comments that should be addressed prior to publication acceptance.

1. The WB data on isolated exosomes is not very convincing especially comparing the results to Figure 8 for example.  A clearer blot with less background would be much better.  The methods do not describe how each lane was normalized to ensure the same amount of material was loaded on each lane. I understand why Beta-actin cant used as a loading  marker with exosomes. The question is how did the authors ensure equal loading in these lanes?

2. Figures 9 and 10: The authors need to determine the percent difference of PDL1 expression between lanes using some method densitometer determination. This result should be normalized to the B-actin lane.

3.  Figure 11. A higher magnification of the IHC staining is required to confirm that the brown stain is truly in the cells and not background.

4. Figure 12. Panel A.  The authors do not explain the presence of PD-L1 at day 2 that disappears days 3 and 4 and reappears day 5.

Author Response

Reviewer 2

Comment 1

  1. The WB data on isolated exosomes is not very convincing especially comparing the results to Figure 8 for example.  A clearer blot with less background would be much better.  The methods do not describe how each lane was normalized to ensure the same amount of material was loaded on each lane. I understand why Beta-actin cant used as a loading marker with exosomes. The question is how did the authors ensure equal loading in these lanes?

Response 1

Thanks for the reviewer’s critical comments. 143B and HT1080 cells grown to 70% confluence were washed twice with PBS and then grown in serum-free MEM medium. The culture medium was used for ultracentrifugation and resuspended in 10 μL of PBS. Thus, all were unified and extracted at the same enrichment rate. This content was re-described in ‘Exosome isolation’ of  Materials and methods. The following is the text of the document. The red text is the revised part.

Exosome isolation

Exosomes were extracted in vitro by ultracentrifugation from the culture medium (20). 143B and HT1080 cells grown to 70% confluence were washed twice with phosphate-buffered saline (PBS) and then grown in 25 ml serum-free MEM medium. After 48 h of incubation, the conditioned medium was collected and centrifuged at 3000 × g for 15 min at 4 °C to thoroughly remove cellular debris. The supernatants were ultracentrifuged at 100,000 × g for 70 min at 4 °C. The pellets were washed with PBS, ultracentrifuged, and resuspended in 10μL PBS. In vivo, exosomes were extracted from serum using blood collected from mice. Serum samples were ultracentrifuged using the same method. Isolated exosomes were stored at –80 °C until needed.

Comment 2

  1. Figures 9 and 10: The authors need to determine the percent difference of PDL1 expression between lanes using some method densitometer determination. This result should be normalized to the B-actin lane.

Response 2

Thank you for your suggestion. We added band intensity data using Image J software in Figure9,10,12. This content was re-described in ‘Western blot analysis’ of Materials and methods. The following is the text of the document. The red text is the revised part.

Western blot analysis

Whole cell lysates or exosomal proteins were separated by 12% sodium dodecyl sulfate-polyacrylamide gel electrophoresis and transferred to nitrocellulose membranes. Blots were blocked using 5% nonfat dry milk for 1 h at room temperature and incubated overnight at 4 °C with the corresponding primary antibody at the dilution recommended by the supplier, followed by incubation with an HRP-labeled secondary antibody (Cell Signaling Technology) for 1 h at room temperature. Blots on the membrane were developed using ECL detection reagent (Pierce). CD9 (Santa Cruz Biotechnology, cat. sc-9148), CD63 (Santa Cruz Biotechnology, cat. sc-15363), and CD81 (Santa Cruz Biotechnology, cat. sc-9158) were used as exosome markers, and actin (Santa Cruz Biotechnology, cat. sc-47778) was used as the loading control. The intensity of each band was compared using Image J software.

Comment 3

  1. Figure 11. A higher magnification of the IHC staining is required to confirm that the brown stain is truly in the cells and not background.

Response 3

Thank you for your pointing. We added enlarged photos with IHC in Figure11

Comment 4

  1. Figure 12. Panel A.  The authors do not explain the presence of PD-L1 at day 2 that disappears days 3 and 4 and reappears day 5.

Response ï¼”

We agree that the result of Figure 12 Panel A is false positive. We changed WB data from another result of Figure 12 Panel A.

Round 2

Reviewer 1 Report

Comments and Suggestions for Authors

ok for my comments 1 and 3. 

Regarding my comment 2, maybe I was not clear enough.

The quality of the WB pictures is not at a good scientific level.

It is clearly not good enough in Fig 2 and Fig 3 to give accurate conclusion.

An example: concerning CD63 WB, the bands are very close to the 46 MW marker (Fig2) but closer to the 31 MW marker in Fig 3, showing clearly a mistake of interpretation.

Please provide better pictures for these analyses or remove these figures

In Fig 9, 11 and 12, the authors described expression modifications (increases or decreases). As no quantification with statistical test was performed, only expressions tendency can be explained....

Please give a quantification of the WB bands according to WB replicates.

With 1 picture and no quantification, it can be only preliminary data. Modify the text and your conclusions accordingly.

Author Response

I am very sorry for the delay in replying.

Comment 1

It is clearly not good enough in Fig 2 and Fig 3 to give accurate conclusion.

An example: concerning CD63 WB, the bands are very close to the 46 MW marker (Fig2) but closer to the 31 MW marker in Fig 3, showing clearly a mistake of interpretation.

Please provide better pictures for these analyses or remove these figures

Response 1

Thank you for your pointing. Concerning CD63 WB, we consider that the band in Figure 2 is not 40 kDa. Therefore, we think that the exosomal markers for HT1080 and 143B in Fig. 2 are bands at CD9 and 81. We revised the figures and papers (2 and 3 line of ‘Results’ section).

In HT1080 cells, exosome markers CD9 and CD81 were detected, and PD-L1 was highly expressed (Fig. 2A). In 143B cells, exosome markers, CD9 and CD81 were detected (Fig. 2B).

Comment 2

In Fig 9, 11 and 12, the authors described expression modifications (increases or decreases). As no quantification with statistical test was performed, only expressions tendency can be explained....

Please give a quantification of the WB bands according to WB replicates.

With 1 picture and no quantification, it can be only preliminary data. Modify the text and your conclusions accordingly.

Response 2

Thank you for your pointing. We have only one data set for each of Figures 9, 11, and 12. We do not have enough time to perform another experiment after purchasing new mouse. I am sorry I can not listen to your suggestion.

Round 3

Reviewer 1 Report

Comments and Suggestions for Authors

ok for comment 1

concerning comment 2:

it is not possible to describe "increase" and "decrease" expression of proteins according to only 1 WB and only qualitative data, with no quantification. The text has to be modified, replace "increase" by "visual tendency to increase" and "decrease" by "tendency to decrease"

Author Response

Comment 1

it is not possible to describe "increase" and "decrease" expression of proteins according to only 1 WB and only qualitative data, with no quantification. The text has to be modified, replace "increase" by "visual tendency to increase" and "decrease" by "tendency to decrease"

Response 1

Thank you for your pointing. We replaced “increase or decrease” by “tendency to increase or decrease” as pointed out. We revised the “Results” and “Figure legents” sections of the paper.

We revised as follows “Effect of irradiation on exosomal PD-L1 protein levels in vitro” in the “Results” section.

Lines 2-4: When the radiation doses were set to 0, 3, 6, and 9 Gy, 6 Gy irradiation tended to increase the expression of cPD-L1 in HT1080 cells at 24 and 48 h (Fig. 9A).

Lines 4-5: In 143B cells, the cPD-L1 expression level tended to be increased in a radiation intensity-dependent manner 48 h after irradiation (Fig. 9B).

Lines 5-7: In contrast, exosomes obtained from the culture medium of irradiated HT1080 cells showed an intensity-dependent tendency to increase in exosomal PD-L1 protein levels at 48 h after irradiation (Fig. 9C).

Lines 7-9: A similar study of 143B cells showed that exosomal PD-L1 levels tended to be increased 24 h after 6-Gy irradiation (Fig. 9D).

We revised as follows “Effect of irradiation on exosomal PD-L1 protein levels in vivo” in the “Results” section.

Lines 5-7: In tumor tissues, cellular PD-L1 expression tended to be upregulated two days after irradiation, followed by a tendency to decrease in PD-L1 protein levels (Fig. 11A).

Lines 9-11: In HT1080 cells, the expression of exosomal PD-L1 detected by anti-human PD-L1 antibody strongly tended to be enhanced five days after irradiation.

Lines 11-12: In contrast, the expression of exosomal PD-L1 detected by anti-mouse PD-L1 antibody strongly tended to be enhanced three days after irradiation and decreased thereafter (Figs. 12A, B).

Lines 12-14: In 143B cells, human and mouse-derived exosomal PD-L1 expressions strongly tended to be enhanced one day after irradiation (Fig. 12C, D).

We revised as follows “Figure 9,11 and 12” in the “Figure legends” section.

Figure 9 Effects of irradiation on PD-L1 expression in vitro

HT1080 (A) and 143B (B) cells were irradiated, and changes in cellular PD-L1 levels were detected by western blotting. Exosomes isolated from the media of HT1080 cells (C) and 143B (D) cells were used to examine PD-L1 expression after radiation. In HT1080 cells, 6-Gy irradiation tends to increase the expression of cPD-L1 at 24 and 48 h. An intensity-dependent tendency to increase in exosomal PD-L1 protein levels is observed at 48 h after irradiation. In 143B cells, the expression level of cPD-L1 tends to be increased in a radiation intensity-dependent manner 48 h after irradiation. Exosomal PD-L1 levels tend to be increased 24 h after 6-Gy irradiation.

Figure 11. Effects of irradiation on PD-L1 expression on immunohistochemistry

Immunohistochemical findings of implanted tumor using anti-human PD-L1 antibody after radiation are shown. In HT1080 cells, cellular PD-L1 expression tends to be upregulated two days after irradiation (A). Irradiation to 143B tumor cells does not show histological upregulation of cellular PD-L1 (B).

Figure 12. Effect of irradiation on the expression level of PD-L1 in individual mice

Exosomes were isolated from mice transplanted with HT1080 cells (A, B) and 143 B cells (C, D), and western blotting was performed using an anti-human or mouse PD-L1 antibody.

In HT1080, human-derived exosomal PD-L1 expression strongly tends to be enhanced five days after irradiation. In contrast, mouse-derived PD-L1 expression strongly tends to be enhanced three days after irradiation and decreased thereafter. In 143B cells, exosomal PD-L1 from humans or mice strongly tends to be enhanced one day after irradiation.

Round 4

Reviewer 1 Report

Comments and Suggestions for Authors

thank you for the last revision

Author Response

Thank you for your approval.